# Cracking the neural code for word recognition in convolutional neural networks

**Aakash Agrawal**[1]*, **Stanislas Dehaene**[1,2]

**1** Cognitive Neuroimaging Unit, CEA, INSERM U 992, Université Paris-Saclay, NeuroSpin center, Gif/Yvette, France, **2** Collège de France, Université Paris Sciences Lettres (PSL), Paris, France

* aakash@alum.iisc.ac.in

**Data Availability Statement:** All data and code necessary to reproduce the results are available in an Open Science Framework repository at https://osf.io/j5nvs/.

## Abstract

Learning to read places a strong challenge on the visual system. Years of expertise lead to a remarkable capacity to separate similar letters and encode their relative positions, thus distinguishing words such as FORM and FROM, invariantly over a large range of positions, sizes and fonts. How neural circuits achieve invariant word recognition remains unknown. Here, we address this issue by recycling deep neural network models initially trained for image recognition. We retrain them to recognize written words and then analyze how reading-specialized units emerge and operate across the successive layers. With literacy, a small subset of units becomes specialized for word recognition in the learned script, similar to the visual word form area (VWFA) in the human brain. We show that these units are sensitive to specific letter identities and their ordinal position from the left or the right of a word. The transition from retinotopic to ordinal position coding is achieved by a hierarchy of "space bigram" unit that detect the position of a letter relative to a blank space and that pool across low- and high-frequency-sensitive units from early layers of the network. The proposed scheme provides a plausible neural code for written words in the VWFA, and leads to predictions for reading behavior, error patterns, and the neurophysiology of reading.

## Author summary

Reading is a fundamental skill in modern society, yet the neural mechanisms that allow us to quickly recognize words remain poorly understood. Our research aims to unravel how the brain achieves invariant word recognition—the ability to recognize words regardless of their position, size, or font. We studied artificial neural networks trained to recognize words, mirroring human learning. Our findings reveal that these networks develop specialized units for word recognition, similar to the Visual Word Form Area in the human brain. These units are sensitive to specific letters and their positions within a word. Crucially, we discovered that they achieve this by detecting the spaces around words as reference points. This creates a hierarchical system where early layers detect basic features and spaces, while higher layers combine this information to recognize specific letters at certain positions relative to word edges. This "space bigram" model reconciles previous theories of letter bigrams and letter-position coding. Our results suggest that most written

**Funding:** A.A. is supported by the Paris Region fellowship Programme as a part of the Horizon 2020 program under the MSCA no. 945298. S.D. is supported by the "TOPLEX" ANR program (ANR-20-CE37-0002). The funders had no role in study design, data collection and analysis, decision to publish, or preparation of the manuscript.

**Competing interests:** The authors have declared that no competing interests exist.

languages may be processed using similar basic principles. This understanding could inform better methods for teaching reading and treating reading disorders.

## Introduction

Distinguishing two visually similar objects, regardless of huge variations in location, size, illumination and other irrelevant factors, is a problem that the human ventral visual pathway solves with remarkable efficiency. This capacity for invariant visual recognition is particularly evident in the case of fluent reading. Three or four times per second, fluent readers land their eyes on a word and reliably recognize it among tens of thousands of similar entries in the mental lexicon. A specific challenge raised by fluent reading is the need to distinguish words such as FROM and FORM that only differ in relative letter position [1] and to do so invariantly over a wide range of fonts, sizes, spacings, and retinal positions [2–4]. Here, based on detailed simulations, we propose a precise hypothesis about the neural circuit that solves this problem.

In recent years, the cortical areas underlying fluent reading have begun to be resolved. The acquisition of literacy leads to the formation of a specialized word-responsive region in the ventral visual cortex, the Visual Word Form Area (VWFA) [5–7]. Brain imaging shows that this region comprises patches of cortex that become highly attuned to stimuli in the learned script [8–10] and preferably responds to stimuli that respect the distributional statistics of letters in the learned language [11–14]. The VWFA responds in a largely invariant manner to identical words that vary in case and location across the visual field [15–17], although it remains weakly modulated by absolute position [18]. Simultaneously, the VWFA differentiates anagrams such as RANGE and ANGER that differ only in the order of their letters [17].

Despite those advances, the nature of the neural code underlying this invariant recognition remains unknown. Two broad classes of theories can be opposed [19]. Contextual schemes assume that letters are encoded relative to the location of other letters. For instance, the visual system may extract bigrams, i.e., ordered pairs of letters. Thus, FORM and FROM would be distinguished by their bigrams "OR" versus "RO", regardless of where they occur on the retina. Encoding the most frequent such bigrams would suffice to recognize many words [1,20,21]. Positional schemes, on the other hand, assume that words are encoded by a list of letters, each attached to its relative ordinal location within the word [22–24]. For instance, each letter may bear a neural code for its approximate ordinal number relative to the beginning of the word [see 19 for a list of relative positional schemes].

Functional MRI data initially supported the bigram hypothesis by revealing stronger VWFA responses to stimuli containing frequent letter bigrams [11,12]. However, recently, evidence from psychophysics and time-resolved intracranial recordings suggests that frequent bigrams may not contribute to recognition as much as was initially thought, at least during the first ~300 ms of word recognition, where only frequent letters are separated from rare letters or non-letters [13,19,25]. What develops with literacy is the compositionality of visual word representation, which increases the dissimilarity and independence between individual letters at nearby locations [26,27]. Compared to contextual bigram coding, ordinal letter coding provides a better fit to the psychophysical distance between letter strings [25], the early intracranial responses to written words [13], and the responses of word selective units in artificial deep networks [28].

What remains to be understood is how such an invariant ordinal code is achieved by neural circuits. Most cognitive models of reading simply beg the question by taking as input a bank of position-specific letter detectors, responding for instance to letter R in 2nd ordinal position [22–24], and implicitly assuming that some earlier unknown mechanism normalized the input

for size, font, and retinal position to eventually encode letters in a relative (ordinal) rather than absolute (retinal) spatial reference frame.

Resolving the neural code for reading is difficult, due to the poor resolution of functional magnetic resonance imaging (fMRI) and the lack of animal models for detailed neurophysiological investigations. Although written words have been used as stimuli in behavioral and electrophysiological experiments [29–31], they have not yet led to an elucidation of the neural architecture for reading. Here, we show that precise predictions about the neurophysiological architecture for invariant reading can be obtained by studying convolutional neural networks (CNN) models of the ventral visual cortex. Similar to literate humans, the neural representations of these networks can be recycled for reading. We trained CNNs to recognize written words in different languages and analyzed their responses to both trained and novel scripts. After validating those models against earlier psychophysical studies [26], we investigated how they achieve invariant word recognition by characterizing their units' receptive fields to letters and strings at various stages. This led us to discover a novel principle of relative position coding, "space bigrams", which accounts for existing data and leads to new predictions about the neurophysiology of reading.

## Results

### Invariant word identification

We first trained various instances of CORnet-Z, a CNN whose architecture partially matches the primate ventral visual system [32]. To mimic a child's learning process, we initially trained the literate network to recognize 1000 image categories from the ImageNet dataset (base network), and then training was extended to the same 1000 ImageNet categories plus an additional 1000 written word categories, in different writing systems for different instances of the network (Fig 1A). In total, there were 7 different types of literate networks, each independently trained starting from the same base network (i.e., the ImageNet-trained network). These networks could be either monolingual or bilingual (Fig 1A). In contrast, the training of illiterate networks was restricted to the ImageNet dataset, with an equivalent training duration as the literate networks.

Behaviorally, the networks trained on ImageNet reached accuracy levels that were comparable to the earlier reported values for CORnet-Z (top-1 accuracy = 36.8% ± 0.1). With the introduction of words, performance on ImageNet dropped marginally (top-1 accuracy = 36.4% ± 0.4) while becoming excellent on word recognition across different languages, with test words varying in case, font, location, and size (see methods) (top-1 accuracy = 88.2% ± 0.5 for French, 87.5% ± 0.5 for English, 92.2% ± 0.3 for Chinese, 95.5% ± 0.2 for Telugu, 90.9% ± 0.5 for Malayalam). Networks trained on bilingual stimuli reached accuracy levels comparable to monolingual networks (top-1 accuracy = 88.6% ± 0.7 for the English + French network, and 91.0% ± 0.2 for the English + Chinese network). Higher accuracy with words than with images can be attributed to the limited variations of text on a plain background.

### Emergence of script-specific units

We next tested for the emergence of units specialized for the visual form of words, similar to the human VWFA. Within the non-convolutional, penultimate layer of each network (avgIT), we searched for units selective to a given script over and above other categories such as faces or objects, a contrast similar to fMRI studies of the VWFA (see methods). This layer has properties similar to those of the IT layer, except for higher invariance due to pooling; it does not have any additional parameters. Literacy dramatically enhanced the number of script-selective units in this layer, from a mean of 4.2 units in illiterate networks to 40–100 units (Fig 1B).

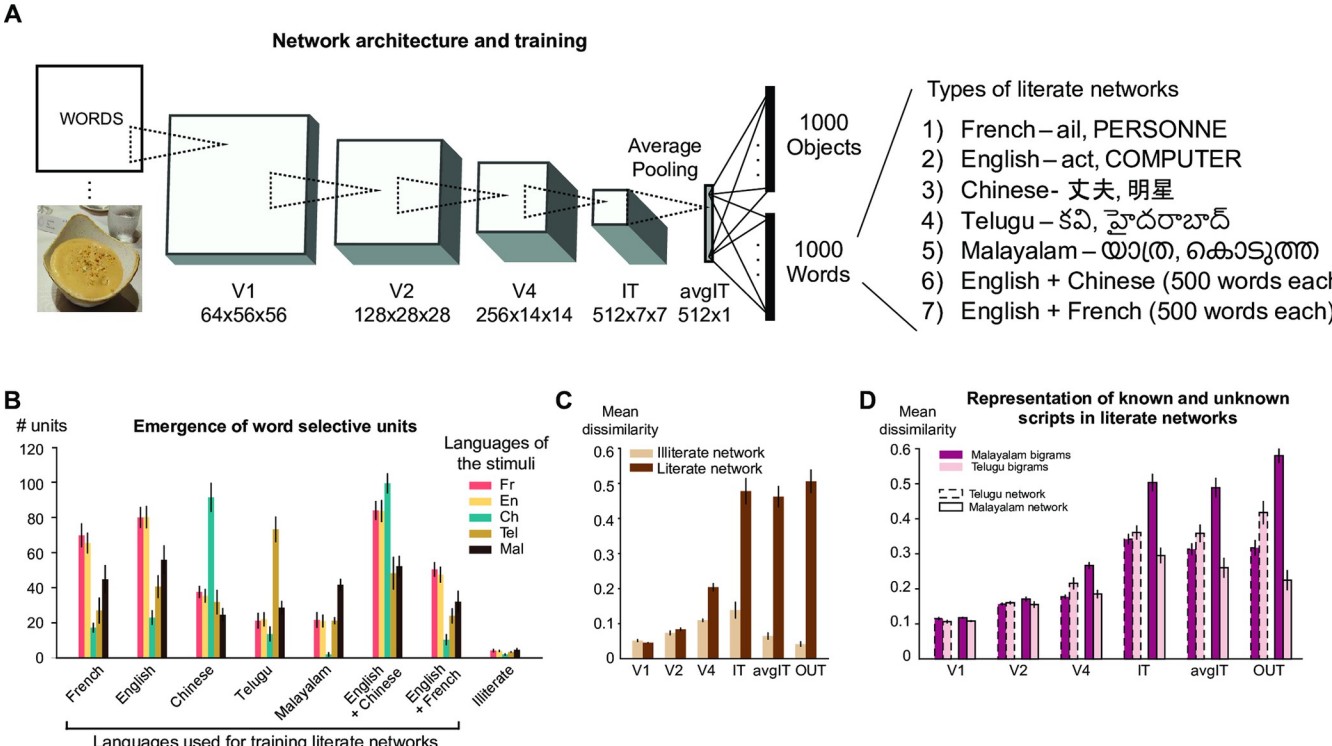

**Fig 1. Properties of a neural network trained to recognize words.** (A). The CORnet-Z architecture was used as a model of the ventral visual pathway. The illiterate network was trained to predict 1000 object categories present in the ImageNet dataset. The literate networks were trained to predict both the 1000 object categories and 1000-word categories from a given language. (B). The number of word-selective units in the avgIT (or avgpool IT) layer of different literate and illiterate networks for both trained and novel scripts. A unit is counted multiple times if they are selective to multiple scripts. Thus, the number of French and English units is identical because they share the same letters. Error bars indicate standard deviation across 5 different instances. (C). Mean dissimilarity estimated across bigrams pairs (n = $^{49}C_2$ = 1176) from different layers of French literate (dark) and illiterate (light) networks (similar results were obtained in networks trained with other languages). The difference between the two networks was highly significant (p<0.0005) starting from the V4 layer. (D). Mean dissimilarity estimated across Telugu (n = $^{25}C_2$ = 300) and Malayalam (n = $^{25}C_2$) bigram pairs from different layers of Telugu (dashed border) and Malayalam (solid border) networks. The difference between the networks for both languages was highly significant (p<0.0005) starting from the V4 layer.

Relative to our previous work [28], where a single script was probed, here we could evaluate the selectivity to the trained script relative to others. In agreement with human fMRI studies of the VWFA [8], many more units responded to the trained script than to untrained ones (Fig 1B). Nevertheless, in literate relative to the illiterate networks, greater responses were also seen to untrained scripts. Furthermore, although few, there were units in each network that exhibited selectivity to novel scripts but not the trained script (n = 3 for French network, 10 for English, 6 for Chinese, 3 for Telugu, 5 for Malayalam, 17 for English + Chinese, 5 for English + French). This finding mimics the experimental observation that, while preferring the learned script(s), the VWFA also responds at a lower level to unknown scripts [8–10].

In our simulations, such generalization to unknown scripts depended on their similarity to the trained script. For instance, the English literate network had a greater number of Malayalam selective units compared to Chinese selective units, presumably due to the presence of rounded symbols in both scripts (Fig 1B). Since a unit can be selective to multiple scripts, we observe the same number of units for both English and French languages across all networks–due to shared script, all units responded to both languages. Furthermore, in "bilingual" networks trained to recognize two scripts, consistent with fMRI of bilingual readers [14], a large proportion of word-selective units were common to the two trained languages, yet more so when the two languages shared the same alphabet (English-French network: 99.8% of French

word selective units, and 100% of English units, also responded to the other script) than when they did not (English-Chinese network: only 75% of Chinese word selective units responded to English, and vice-versa for 84% of English units). This is consistent with our previous findings from 7T imaging of English-French, and English-Chinese bilinguals, where language-specific specialized voxels were found in the latter, but not in the former [14]. Furthermore, despite ~100% overlap in word selective units between English and French scripts, it was still possible to train a classifier to separate the two languages, especially in the later layers of the network (Fig A in S1 Text). Again, this is consistent with prior fMRI data [33] and reflects that fact that these literate networks encode word statistics; for instance, an encoding model trained on words shows a gradual reduction in generalizability when tested on frequent quadrigrams, bigrams, and letters [28]. Even baboons can discriminate between words and pseudowords using just the orthographic information [29]. Overall, the evidence indicates that our networks developed partially language-specific orthographic responses.

The variation in the number of word selective units across different networks may potentially be explained based on the properties of training dataset. For instance, the number of English selective units in the bilingual English+French networks is lower than in the monolingual English- or French-alone networks. This surprising finding is an artificial consequence of a design constraint, namely having the same number of output word categories across all networks–thus, bilingual networks were trained with only 500 words in each language, and presumably could do so with fewer units. In reality, however, bilingual individuals would have double the vocabulary size compared to monolinguals, and thus possibly a larger VWFA. On the other hand, the larger number of English word-selective units in the bilingual English + Chinese networks can be explained as the sum of English units in Chinese-only network (due to shared features between English and Chinese) and English selective units in English + French bilingual network (due to English specific features).

## Improvements in neural discriminability

In humans, literacy leads to script-specific behavioral and neural response enhancements at both early and late visual stages [5,7,34,35] and, in particular, increases in the perceived dissimilarity between letters, bigrams and words [26]. To examine whether this effect was present in our simulations, we compared the mean pair-wise neural dissimilarity between 49 bigrams in French literate and illiterate networks (Fig 1C; see methods). From the V4 layer on, the mean dissimilarity was indeed higher for the literate than for the illiterate network, indicating that the neural population had improved its representation of letter combinations. Similar results were obtained using single letter stimuli (n = 26) that were presented at the center (Fig B in S1 Text). We also repeated the 2x2 analysis reported by [26], which compared the mean dissimilarity between Telugu and Malayalam bigrams in readers of either script. Our simulations replicated the finding that visual dissimilarity is higher for the learned script, and localized this effect to mid-visual areas, starting in area V4 (Fig 1D). Thus, the trained neural network developed representations that are consistent with several human studies. The effect of script properties on neural discriminability remains unknown, and is beyond the scope of this study.

For the remainder of this paper, we focused on networks trained to recognize French words. This script has only 26 unique symbols, making it ideal for an initial step in decoding the neural code of word recognition.

## Letter tuning

Using 1000 words with variable length, we replicated our previous observation that a large amount of variance in the activity of word-selective units in the penultimate avgIT layer could

be captured by a letter X position encoding model (see Methods) [28]. Some units cared about a single letter regardless of its position, but most units cared about one or several letters at a specific ordinal position (Fig 2A). A model-based comparison of various letter-position schemes revealed improved fits when units were assumed to fire at a fixed ordinal position relative to word beginning or ending, rather than relative to either alone or to word center (Fig 2B), in agreement with human behavior [19]. Position tuning was sharper near those edge letter positions (i.e., 1$^{st}$, 2$^{nd}$, penultimate, or end position) compared to middle letter positions (Fig 2C). These findings align with actual recordings of number neurons [36] and with previous behavioral studies that found a greater sensitivity of human readers to detect changes in edge letters than in middle ones [1,25].

## Emergence of letter- and position-invariant units

While the presence of units selective to a given letter independent of their position can easily be explained as a consequence of network architecture (convolution + pooling), it is unclear how these networks developed ordinal position coding units. Since the initial layers have smaller receptive fields, it is very unlikely that their units encode ordinal position. Thus, we hypothesized that ordinal coding arose progressively across the successive layers of the network, as an emerging property of the complexification and broadening of each unit's receptive field. To dissect the reading circuit, we first identified, at each layer of the French literate network, the units preferring words over other stimuli such as faces or objects (see methods). The proportion of word selective units increased with successive layers (percentage of units = 0.02% in V1, 0.6% in V2, 1.76% in V4, 3.61% in IT, 13.6% in avgIT). To evaluate the units' receptive fields, we next tested each word-selective unit with stimuli designed to dissociate retinotopic letter position, word position, and ordinal letter position codes (Fig 3). First, for each unit, we identified its most and least preferred letter (see methods). These letters were then used to create 4-letter stimuli where a single preferred letter (e.g., o) was embedded at various locations within a string of non-preferred letters (e.g., xoxx). In our 5x4 factorial design, retinotopic word position (5 levels) varied across the rows, while ordinal position of the preferred letter (4 levels) varied across the columns (Fig 3A). Across that 5x4 stimulus matrix, units coding for a fixed retinotopic letter position should exhibit a diagonal response profile (Fig 3B-left). Conversely, units that encode a fixed ordinal position, regardless of word position, will have a vertical response profile (Fig 3B-right). We used an image-processing based approach to systematically compute these profiles within each layer (see methods).

In our networks, we indeed observed units that were sharply tuned to either retinotopic or ordinal positions (Fig 3C and 3D). The proportion of retinotopic units was highest in the initial layers of the network (n = 100% in V1, 46.5% in V2, 56% in V4, 20.8% in IT, and 3.4% in avgIT layer). As expected, units in the V1 layer responded to their preferred letter(s) only when presented at a specific retinotopic position. Furthermore, in each layer, we continued to find units sensitive to a given letter regardless of its position in the word, and with a receptive field size increasing along the successive layers of the network. Mid-layer units spanned 2–4 letter positions, and the avgIT units showed complete position invariance (Fig 3C). While such units encode letter identity, they are insensitive to ordinal position and therefore unable to encode relative order or to separate anagrams.

Crucially, however, and consistent with our encoding models, the later layers of the network also had a high proportion of ordinal position coding units (83.1% in the avgIT layer). Interestingly, the units in the intermediate layers also encoded ordinal position, albeit with a smaller receptive field. These units responded only when the preferred letter was presented within the given receptive field and at a fixed ordinal position, often either the first or the last

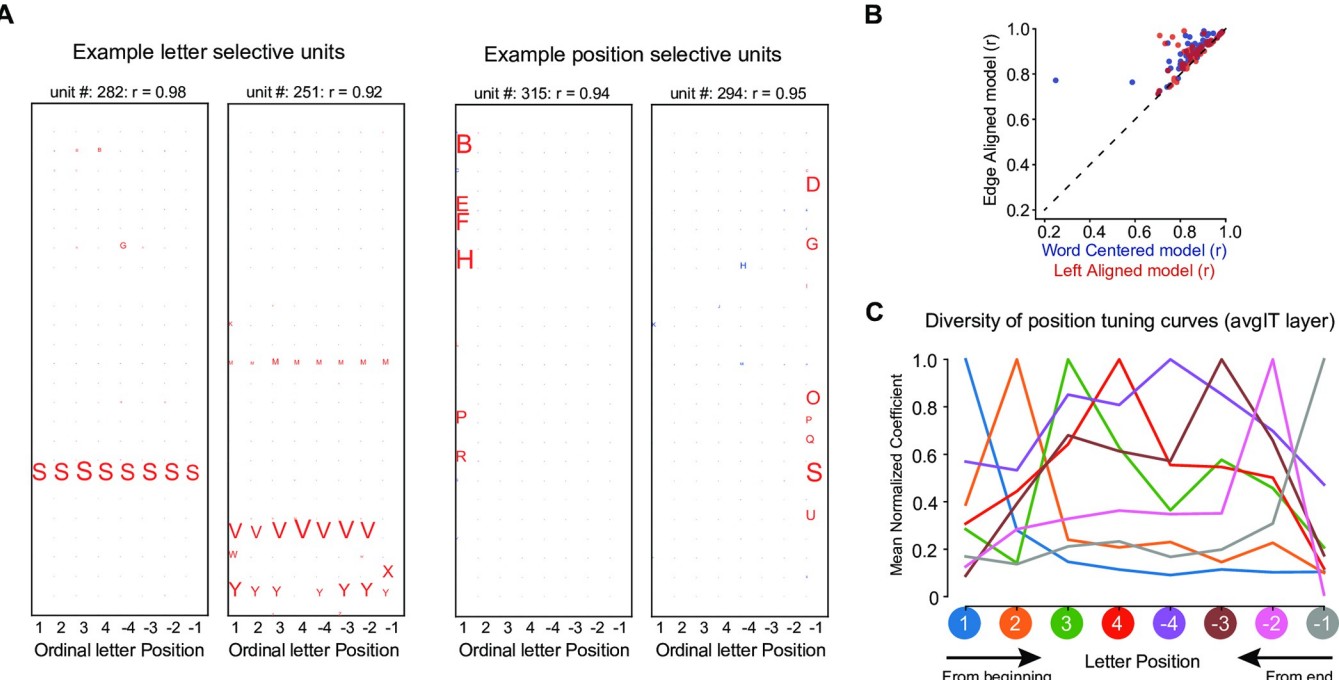

**Fig 2. Position encoding schemes.** (A). Visualization of the letter model coefficients for a few example units. A unit is categorized as letter selective if it responds to stimuli containing a specific letter invariantly across all positions. Similarly, a unit is categorized as position selective if it responds to stimuli containing one or more preferred letters at a specific position. The size of the letter indicates the coefficient magnitude. (B). Comparison between the model fits across all word selective units in the avgIT layer using an edge-aligned position coding scheme versus either word-centered (blue) or left-aligned position coding schemes (red). The dashed line represents the unity slope line. (C). Average coefficients of the ordinal-position regression model in the avgIT layer. Following [36], units were sorted according to their preferred ordinal location from word beginning or ending (colors), and the coefficients of each unit were normalized by dividing by their maximum value and then averaging across units.

position in a word (n = 7.8% in V2, 11.8% in V4, 37.3% in IT). Such units are referred as "edge coding units" and were observed in all layers but V1 (Fig 3D). Across V2, V4, and IT, these units exhibited a broadening of their retinotopic receptive fields, until a complete invariance to retinotopic position and a pure selectivity to ordinal position were attained in the avgIT layer (Fig 3D).

Thus, ordinal coding was achieved by pooling over a hierarchy of edge-sensitive letter detectors. While this scheme was dominant, we also observed units with mixed selectivity that responded to a diverse range of positions that were neither purely retinotopic nor ordinal (Fig C in S1 Text).

Next, we investigated the mechanistic origins of those crucial edge-coding units sensitive to ordinal position. We hypothesized that, within the convolutional layers, units managed to encode approximate ordinal position because their convolution field was jointly sensitive to (1) one or several specific letter shapes, *and* (2) the presence of a blank space (absence of any letter), either to the left (for units coding ordinal position relative to word beginning) or to its right (for end-coding units). We term these units "space bigrams" because they are sensitive to a pair of characters, one of which is a space. The space bigram coding scheme is therefore an extension of the previous open-bigram hypothesis [1,20], which assumed that units would be sensitive to ordered letter pairs such as "O left of R". The only difference (and yet a crucial one) is that it allows one of the two letters to be a space, thus encoding for instance "O left of a space". Indeed, this new scheme makes sense given that space is the most frequent character (~20%) in English, French, and probably all similar alphabetic codes.

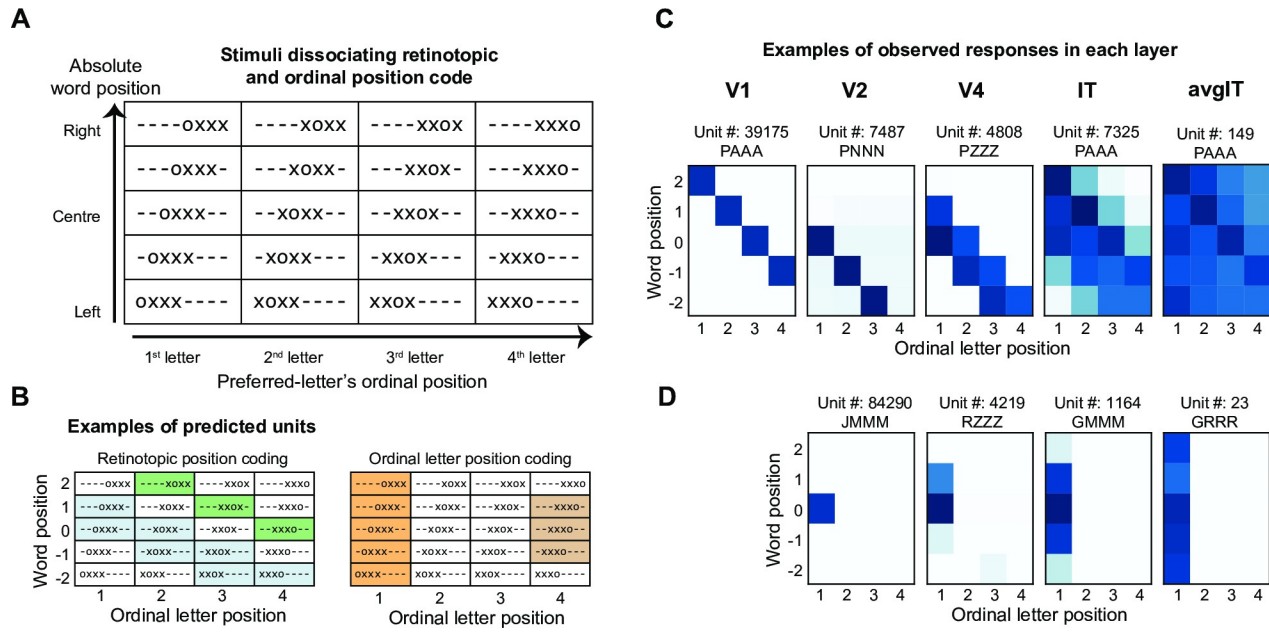

**Fig 3. Transition from absolute (retinotopic) coding to ordinal position coding. (A)**. Schematic of the stimuli used to dissociate absolute vs relative coding. We presented a single preferred letter within a "word" made of multiple non-preferred letters. Here, the preferred letter is 'o', and the unpreferred letter is 'x'. Absolute word position varies across rows, while ordinal preferred-letter position within a string varies across columns. '-' represents blank space. (B). Expected response profile for two types of units with (1) absolute, retinotopic letter position coding (left); and (2) relative, ordinal letter-position coding (right). Each color represents a different unit. (C). Exemplar units from each layer of the French literate network. The unit-id and the string comprising preferred and unpreferred letter is displayed on the top. Darker shades represent a higher response. (D). Same as (C) but for units showing ordinal position coding.

To separate space coding from ordinal position coding, we tested our network's responses to a modified stimulus set where a blank space was introduced between the letters (e.g., x o x). Behavioral and brain-imaging tests show that inserting a single s p a c e between letters does not disrupt normal reading [3,37] and may even facilitate it for some readers [38,39]. To maintain the overall count of possible letter positions, the number of letters in a given stimulus was reduced to three, consisting of one preferred and two non-preferred letters. This resulted in a 4x3 stimulus matrix, where absolute word position (4 levels) varied across rows, and the

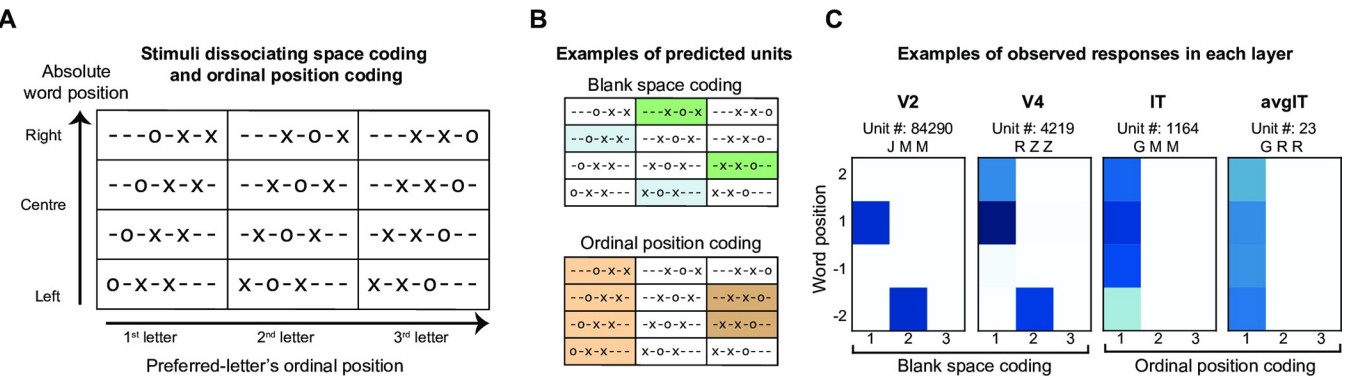

**Fig 4. Transition from blank space to ordinal position coding. (A)**. Same as Fig 3 but with stimuli used to dissociate blank space vs ordinal position coding units. (B). Expected response profile of units with blank space coding (top) and ordinal letter position coding (bottom). Each color represents a different unit. (C). Exemplar units from each layer of the French literate network. The unit-id and the string comprising preferred and unpreferred letter is displayed on the top. Darker shades represent a higher response.

ordinal position of the preferred letter varied across the columns (3 levels; Fig 4A). If edge-coding units were indeed encoding the presence of a nearby blank space rather than a genuine ordinal position, their responses would persist even when the preferred letter was present at the middle location, provided it was flanked by a blank space (Fig 4B-top). Conversely, a genuine ordinal position unit would continue to respond to the same ordinal location within the overall word, even in the spaced condition (Fig 4B-bottom).

Fig 4C shows the responses of the same units as in Fig 3D when tested using the spaced stimulus set. We found a clear transition across layers: V2 and V4 units coded for blank spaces, while IT and avgIT layers encoded ordinal positions. For instance, a specific unit (# 84290) in the V2 layer exhibited a maximal response when its preferred letter 'J' was presented at the beginning of the word, with a receptive field centered on the third retinotopic letter position (Fig 3D). When tested with spaced stimuli, the unit maintained its receptive field location (i.e., third retinotopic position) and continued to respond to the letter J, but now did so whenever it was preceded by a blank space, even if it was not located at the beginning of the word. Units in the V4 layer exhibited a comparable response pattern. However, in the IT layer, units maintained their preference for the first ordinal position even within s p a c e d strings. These findings provide robust evidence that edge coding units initially encode blank spaces and contribute to the ultimate extraction of ordinal position coding.

### Emergence of ordinal position coding units

We hypothesized that ordinal position coding units acquire their sensitivity to ordinal position within the word by pooling over several blank-space units from the preceding layers. To investigate this hypothesis in detail, we examined the input connection profile of IT units where we observed the first evidence for ordinal position coding (Fig 4). Our goal was to characterize, mechanistically, how each IT unit acquired its selectivity by visualizing its strongest V4 input units and understanding, in turn, how the cells were tuned (see Methods). Such a dissection is illustrated in Fig 5 for one ordinal IT unit. We observed that each IT cell receives two types of V4 inputs: (1) from V4 cells that are already highly selective to one or a few letters, but with a broad retinotopic receptive field; and (2) from V4 cells that typically have broader selectivity for several letter identities, but also care about the presence of a blank space, either to their left or to their right (Fig 5B). To further probe this, we examined the responses of these V4 units to single letters presented independently at each of the 8 spatial positions. This confirmed that units with retinotopic coding exhibit preferential responses to a few letters within their receptive field, while blank-space coding units respond to a broader range of letters (Fig 5C). These IT units also received inhibitory inputs from similar types of V4 units but with different letter selectivity. This led IT cells to exhibit a selectivity to a narrower set of letters, despite receiving excitatory inputs over a broad range of letters. Fig D in S1 Text shows similar results for another IT unit encoding the last letter. This unit received both excitatory and inhibitory inputs from units that encoded a diverse, yet distinct, range of letters followed by a blank space. The combination of these inputs fine-tuned the letter selectivity of the IT unit. Additionally, the IT unit received excitatory input from a retinotopic V4 unit selective to the IT unit's preferred letter, further refining its response properties. This mechanism illustrates how the interplay of broad inputs and specific inhibitory connections can shape the precise tuning of higher-level visual word recognition units.

Interestingly, IT units that preferred medial ordinal positions exhibited a distinct connectivity pattern. These units received excitatory input from retinotopic units, which provided information about specific letter features at various locations. Crucially, they also received inhibitory inputs from edge-coding blank space-bigram units. This inhibitory input is

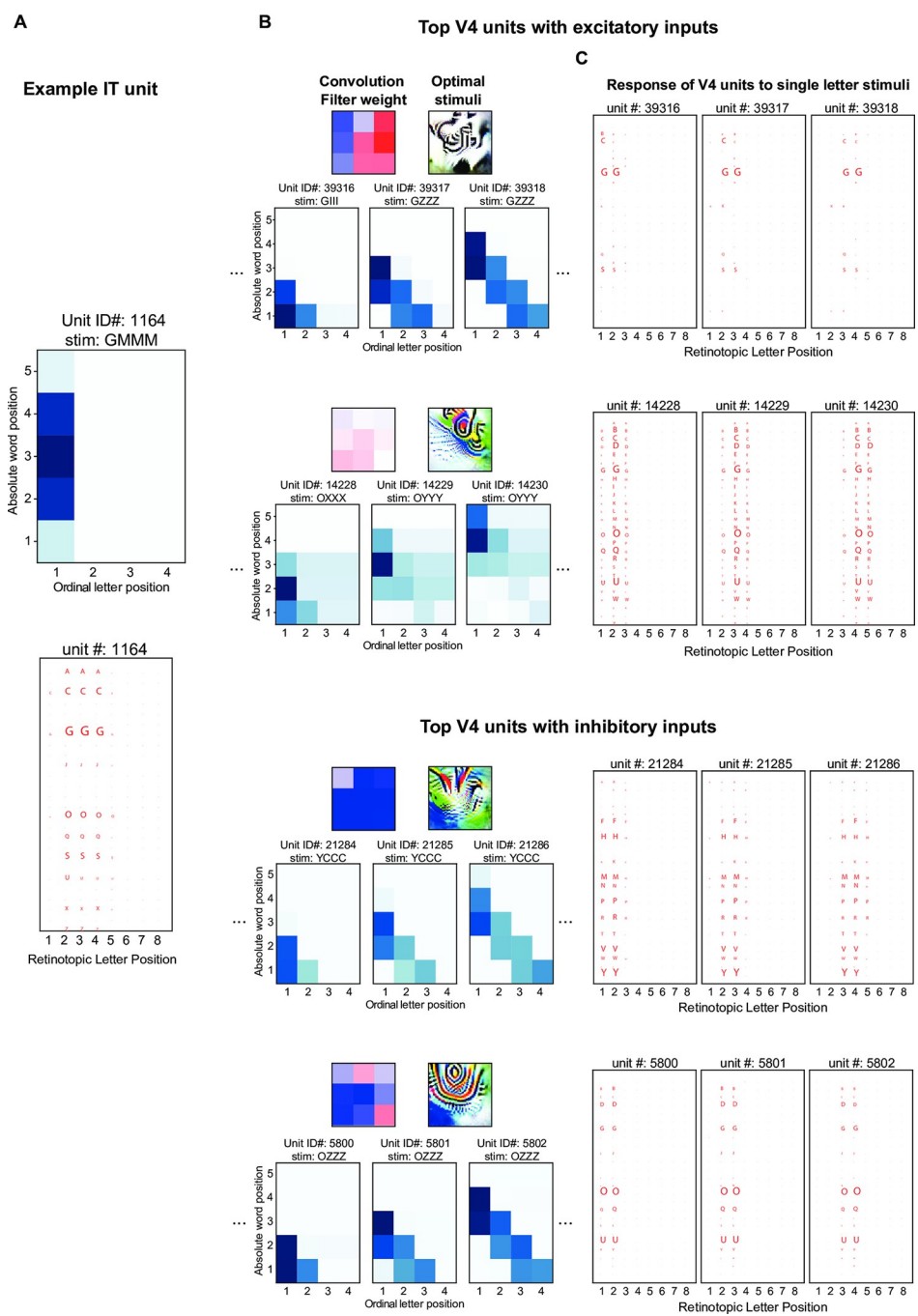

**Fig 5. Functional connectivity between V4 and IT units explains the emergence of ordinal position coding.** (A). Response profile of an example IT unit that encodes ordinal position. The unit is most responsive to letter G in first position. Normalized response profiles to single letters stimuli presented at different spatial locations suggest that this unit is weakly selective to other curved letters as well. (B). Response profile of word-selective units in the V4 layer to their preferred stimuli that lie within the receptive field of the specific IT unit. Two V4 channels with excitatory inputs (top) and two with inhibitory inputs (bottom) to that IT unit are shown. For each channel, we visualized the convolution filter weights that connect V4 and IT layers and generated the stimuli that would maximally activate a given filter using the activation-maximization method. Additionally, we visualized the response profile of three sample units. For example, the 3 units on the top row are highly sensitive to letter G regardless of where it appears. The 3 units on the bottom row are sensitive to several round letters including G and O (see panel C), but only if they appear right of a space. (C). Normalized response profile of the units shown in (B) to single letters stimuli presented at different spatial locations.

significant because it effectively suppresses responses to letters at the beginning or end of words, allowing these IT units to specialize in detecting letters in middle positions. Thus, the edge positions, which are already encoded in V4, serve as important anchors in the word recognition process. These edge-coding units help shape the responses to other medial letter positions in IT by providing a frame of reference for letter position within a word. Fig E in S1 Text illustrates this connectivity pattern and its functional consequences. It's worth noting that we observed similar organizational principles when investigating the connectivity between V4 and V2, suggesting that this hierarchical refinement of letter position coding is a consistent feature across multiple levels of the ventral visual stream (see Section B in S1 Text for detailed analysis).

Using similar approaches, we also investigated the emergence of blank-space coding and retinotopic letter coding in V2 layer (Section C in S1 Text). Interestingly, the output of low-frequency filters in V1 layer primarily modulated the responses of blank-space units in V2, thus giving them a high sensitivity to the edges of words, but a low sensitivity to specific letter; and, conversely, the output of high-frequency filters primarily influenced the response of retinotopic units in V2, thus giving them a fine sensitivity to specific letter shapes. Overall, we conclude that the sensitivity to specific letters at a given ordinal position emerged progressively across layers, through the pooling of units at the immediately preceding layer that are (1) less spatially invariant, but (2) already sensitive to some letters and the presence of a space at some distance to the left or right. The pooling mechanism simultaneously abstracts away from retinotopic information while refining each unit's letter specificity and ordinal coding.

## Optimal stimuli for each layer

To study the properties of neural networks, a complementary approach consists of identifying the most and the least preferred stimuli for individual units. This investigation can be performed using Feature Visualization [40], which is based on image-level gradient descent. By iteratively adjusting the pixel values of an image, a stimulus is generated that maximizes/minimizes the activation of a specific unit. Analyzing these generated images enables us to identify the distinguishing features preferred by each unit.

Here, we started with 3 French words of varying length (AIR, PAIN, and SQUARE) and found the unit with the highest activation in each layer. Next, for each unit, we generated its most preferred stimuli using the network interpretability toolbox (see methods). In the early layers (V1 and V2), units preferred dark-oriented lines and curved shapes against a white background (Fig 6), presumably reflecting the background of the training word dataset. Consistent with earlier reports [41], feature complexity and receptive field size increased in later layers of the network. Remarkably, from IT and output, the automated image optimization process recovered word fragments that partially matched the word identity originally used to select the units (Fig 6). Furthermore, the optimal stimuli repeated those fragments over the visual scene, with an inter-stimulus spacing of at least one letter, thus reflecting the large receptive fields, translation invariance, and abstraction of features in deep layers of CNNs.

## Discussion

Most cognitive models of reading assume a letter by position code as input [24], yet without showing how this information might be extracted from a page of text. Here, we dissected literacy-trained convolutional neural networks (CNNs) and formulated a precise hypothesis about how letters and their positions are extracted from visual strings. In CNNs, literacy training led to the emergence of units with selectivity for words, particularly in the trained script relative to untrained scripts. This finding is akin to the formation of a Visual Word Form Area (VWFA)

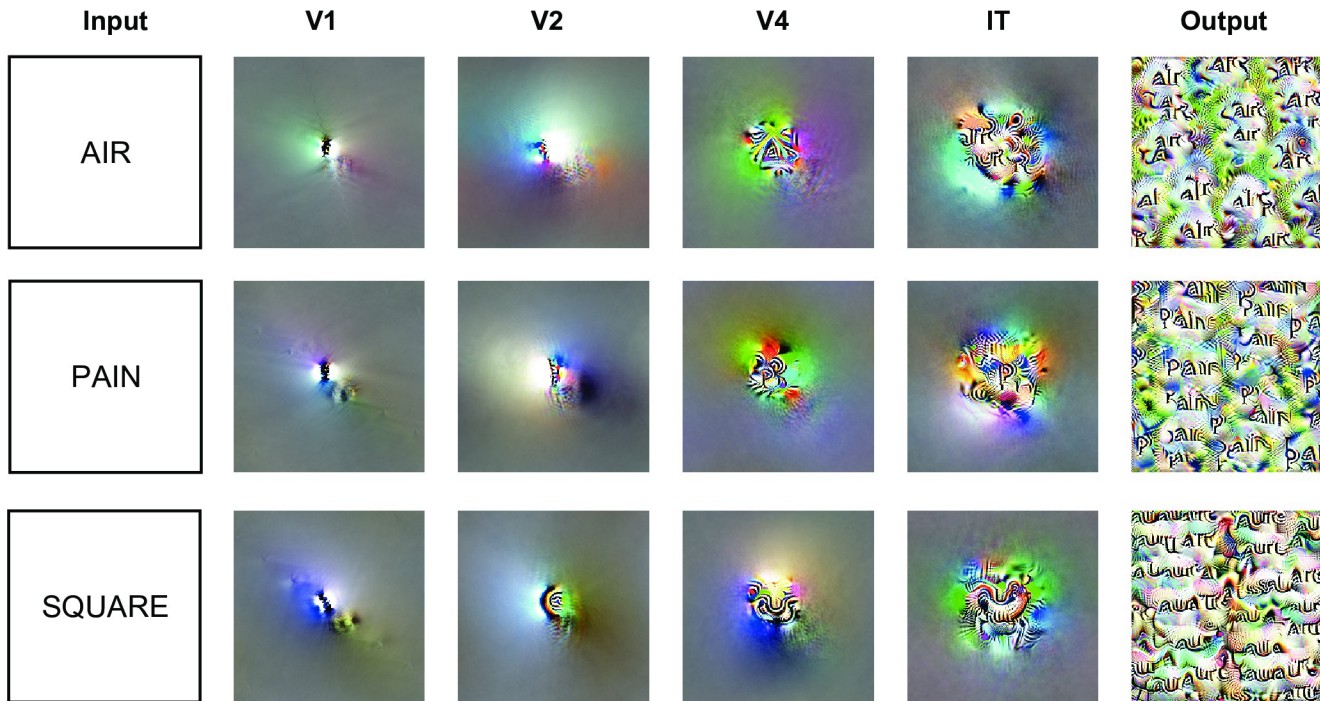

**Fig 6. Activation-maximization of word selective units.** For each French word (input), the features of the channel whose units evoked the highest response within a given layer are shown. For visualization purposes, features are displayed at the central location. Since the avgIT layer is formed by the average pooling of IT layer units, its features are identical to the IT layer and are therefore excluded.

[6,8,15]. The main advance here is that we clarify how the firing of these units collectively encodes a written word. We first show that their response profile fits an approximate letter by ordinal position code, with position being encoded as an approximate number of letters relative to either the left or the right word edge. We then used a neuro-physiological approach to identify their preferred letter(s), their response to strings that orthogonally vary letter position, spacing, and word position, and the connections they receive from earlier layers.

The outcome is a precise mechanistic hypothesis about the neural circuitry for invariant word recognition. The model, presented in Fig 7, explains how units in the highest layer progressively acquire a receptive field which is jointly sensitive to a letter and to its ordinal position, by relying on a pyramid of lower-level units in early layers, some of which are sensitive to letters and others to the presence of a space at a certain distance to the left or to the right. This "space-bigram" model can be seen as a reconciliation of two previous proposals: letter bigrams and letter-position coding. In agreement with the letter-bigram model, IT neurons encode frequent character pairs–except that one of those characters is a space (which is indeed the most frequent character in text). As a result, neurons end up responding selectively to a single or a few letters at a given ordinal position from either word beginning or word ending—an approximate ordinal code, compatible with recent psychophysical and intracranial recordings [13,19,25]. This code is extracted by a feedforward hierarchy of neurons with increasingly larger retinotopic receptive fields, jointly sensitive to the high-frequency shapes that make up letters and to the low-frequency patterns that signal word boundaries (Fig 7) [see 42 for a similar finding]. Thus, the ordinal positions of all letters in a word can be extracted in a fast, parallel, feedforward manner, unlike previous models that relied on hypothetical mechanisms of serial left-to-right processing or temporal coding [21,24]. Such parallel processing, in which all letters, regardless of their position, are processed simultaneously, is compatible with the

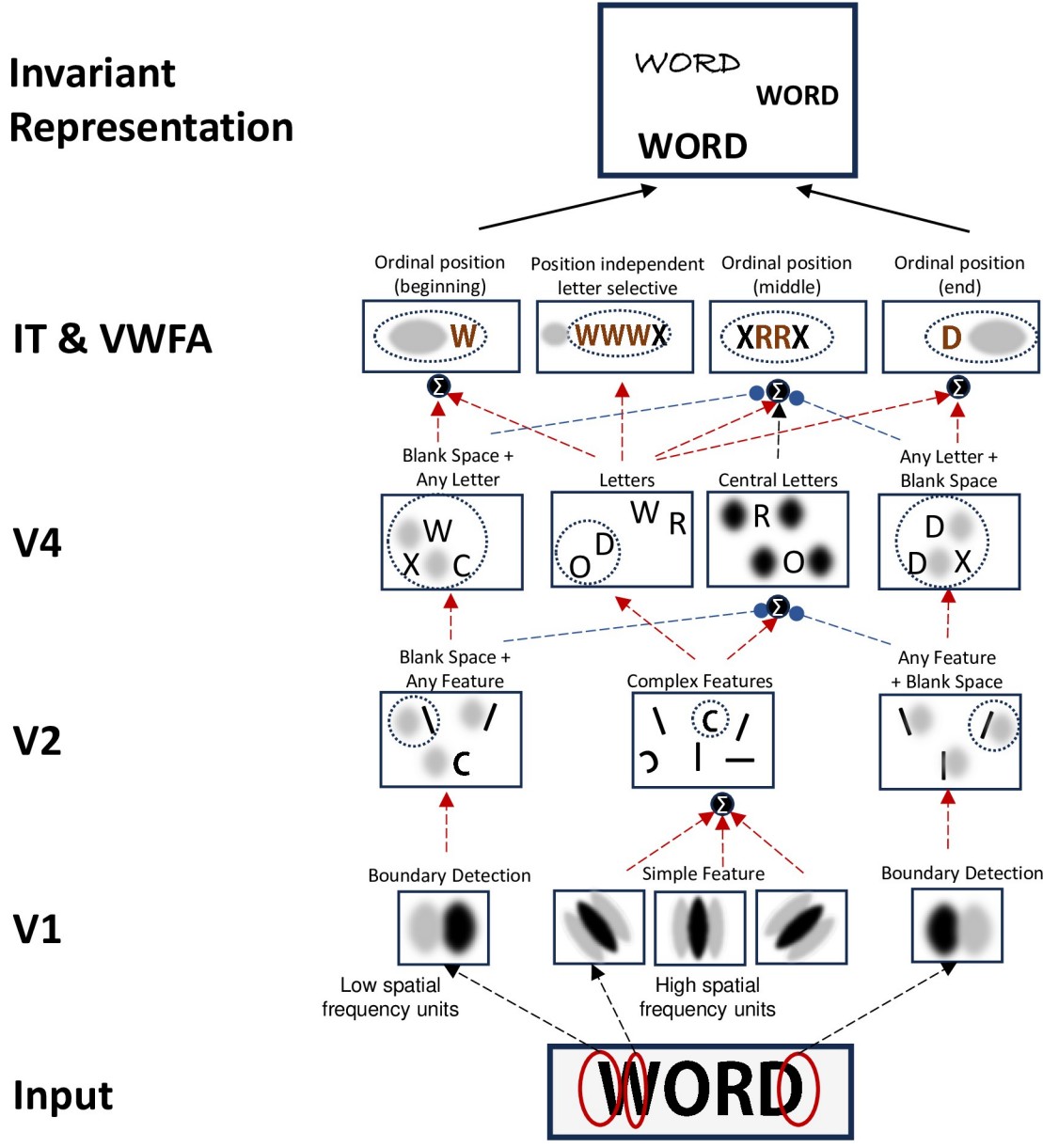

**Fig 7. Schematic model of how an invariant neural code for word recognition is achieved.** The figure illustrates the types of neural tuning observed at successive layers of the ventral visual hierarchy, and how their interconnections (red = excitation, blue = inhibition) shape high-level receptive fields. The initial layers of the network extract features associated with low and high spatial frequencies. High spatial frequency units allow for the progressive extraction of the precise features of letters at specific retinotopic positions, and their pooling leads to an increasingly precise letter code, increasingly invariant for retinotopic location, in downstream regions V2, V4, and IT. Starting in V2, the combination of low spatial frequency units marking word boundaries and letter-tuned units leads to the emergence of edge-coding space bigrams. These respond to a letter or a small set of letters next to a blank space (either to the left or right), thus encoding the first and last letters of a word. In later layers, inhibitory inputs from edge-coding space bigram units lead to the emergence of units that respond only to mid-letter positions. Up to the V4 layer, receptive fields (illustrated by dotted ovals) maintain retinotopic properties. Genuine ordinal position coding begins to emerge in the IT layer, where multiple units combine to form a distinct distributed vector code for individual words. The progression from retinotopic to more abstract representations occur gradually across the network layers, ultimately achieving an invariant neural code for word recognition.

absence of a word length effect in normal reading [37,43,44]. Interestingly, a word length effect emerges suddenly when words are exceedingly rotated or s p a c e d [3,37], i.e. conditions under which the present receptive fields would cease to operate. Neuropsychological and brain-imaging observations indicate that, under such conditions, the ventral visual pathway ceases to be sufficient for reading, and is supplemented by serial attention mechanisms involving posterior parietal cortex [37,45]. In the future, it would be interesting to supplement the present model with a selective attention mechanism and other components needed to simulate slow serial reading.

Our simulation shows that it is possible for IT units to be, simultaneously, highly sensitive to the ordinal position of a letter within a word, and highly insensitive to the overall position of that word in space. Note, however, that this invariance for absolute word position developed progressively across layers, and that even in the highest layers, a few units continued to respond to their preferred letters at specific retinotopic positions, thereby accounting for the ability to decode stimulus position in the VWFA [18]. It should be noted that units selective to a specific letter are likely to respond to a wide range of objects with similar features; for example, a unit selective to the letter 'O' might also respond to an image of a ball. This observation is supported by our finding that units selective to words often respond to multiple scripts. Thus, we do not claim to have identified grandmother cells specific to individual letters. Instead, we propose a vectorial representation of letters, which aligns with previous findings [25]. The responses of individual units to letters, when considered together, form a highly compositional representation of words which is also sparse and can be likened to a "barcode" [28]. Our research suggests that the neural coding of letter strings may be characterized by a distributed and sparse activation pattern, which in turn, further upstream, suffices to excite units highly selective to individual words.

The proposed neural code (Fig 7) can also explain several prior findings in the neuropsychology of reading. Psychophysical studies show that edge letters are crucial for efficient word recognition, while middle letters can be partially displaced or transposed [4]. This phenomenon, popularly known as the Cmabrigde Effect [1], arises here because the letter position detectors exhibit sharp ordinal sensitivity only for edge letters, and show increasingly broader tuning for middle letter positions (see Fig 2C) [46]. More generally, this feature of literate neural network models can account for multiple visual form priming effects [47] and the finding of stronger letter transposition effects in medial positions compared to edge positions [28]. Because the present model is solely trained to categorize words, it cannot capture the variations in the size of the transposed letter effect, which vary with the task demands [48]. However, the proposed coding scheme can also explain the transposition errors of patients with letter-position developmental dyslexia [49], the position-preserving perseveration errors of a patient with acquired alexia [19], and the existence of developmental attentional dyslexia, where readers experience the migration of letters from neighboring words while preserving their ordinal positions [50,51].

A thorough test of the model will require high-density neuronal recordings in human VWFA and connected sites, which is not immediately available. Currently, human intracranial recordings in this region have been primarily limited to coarse-grain electro-corticography [13,52–54]. High-density human single-neuron recordings are increasingly becoming feasible, however [55,56]. Moreover, the stimuli that we designed to investigating artificial neural networks (Fig 3) could be ideal to investigate human reading-related responses, not only in electrophysiology, but also in brain-imaging experiments with fMRI or MEG. For instance, our model predicts a specific pattern of representational similarity at each layer, which could be investigated at the voxel or sensor level even without having access to single units [57,58]. We hope that the present work will stimulate experiments in this field, similar to previous

examples where simulations of neural networks predicted neuronal properties that were later validated experimentally [36, 59,60]. Further studies should also investigate how the neural code of word recognition is modulated across different task contexts that are thought to affect the circuits for reading [22,61] such as the production of phonemes as output, decoding real words vs pseudowords, etc. Specifically, in Semitic languages (like Hebrew, Arabic, etc.) with a complex morphological structure, transposing letters has a stronger detrimental effect on reading [62]. However, the interaction between morphological structure and letter transposition is observed only in tasks involving lexical access. Using a same-different task, classical letter transposition effects were observed in both Hebrew [63] and Arabic [64].Such findings emphasize the need to investigate the properties of our networks in tasks beyond mere image categorization.

Although the insights from our model were primarily derived using French words, we speculate that the findings could easily be extended to different languages. The differences in the observed neural code, if any, would only be driven by input statistics. For example, in Chinese, the basic units are logographs, themselves composed of a large variety of stroke groups, more numerous than letters, and which may occur only in a limited number of specific characters. Thus, the networks trained to recognize Chinese words might have a higher proportion of location-independent stroke-group coding units and a smaller proportion of location-specific blank-space units. Furthermore, Chinese characters are organized in 2 dimensions, different from the linear organization of alphabetic writing, and therefore we might expect to see position-dependent units that care about the position of specific stroke-groups, not only relative to the left and right spaces, but also to the top or bottom or a character. Such differences may eventually explain why slightly distinct ventral occipito-temporal patches of cortex respond to English and Chinese writing in bilinguals [14]. Similarly, in Akshara languages such as Telugu and Malayalam, characters are frequently modified with matras, also known as vowel diacritics, in a combinatorial manner. Thus, Telugu/Malayalam literate networks might develop units specifically encoding these vowel modifiers. Additionally, the Akshara language system is highly nonlinear i.e., matras can be placed on all four sides of the character. Again, this might lead to the formation of units encoding position along the vertical dimension and not just the horizontal one, as observed here. Further studies will focus on analyzing the other literate networks to confirm the predictions mentioned above. However, the generalizability of the proposed model to different languages is restricted here to the mechanisms associated with bottom-up visual word recognition, independent of other further lexical, phonological, or semantic processing. Conducting a meta-analysis of VWFA studies across various languages will aid in testing whether the VWFA shows significant variations in cortical size and internal organization across different languages.

Overall, we delineate the stages of orthographic processing that lead to invariant visual word recognition. Beyond reading, the proposed hierarchical scheme for moving from retinotopic to ordinal-position codes readily extends to the recognition of the configuration of parts within an object, as studied in the macaque monkey [65]. The same mechanism would also allow to encode the position of parts, not only relative to left/right, but also to top/bottom, as required for mathematical or musical notations. Finally, while we focused entirely on the endpoint of learning, the present work could easily be extended to study the developmental emergence of letter position codes in both models and children [6].

## Methods

### Model architecture and training

Among the many available convolutional neural networks that can predict neural responses along the ventral visual pathway, we chose CORnet-Z architecture for two reasons: 1) It has a

modular structure that resembles the stages of processing in the visual cortex (V1, V2, V4, IT, avgpool IT or avgIT, output), 2) It has fewer parameters, thus lowering the training time while achieving high levels of accuracy on synthetic word datasets [28]. Similar to our previous work, we first trained this network on the ImageNet dataset (phase 1), which contains ~1.3 million images across 1000 categories. This was considered an illiterate network, which encodes the visual properties of objects but not text. Next, we extended the number of output nodes to 2000 (1000 images + 1000 words), with full connectivity to avgIT layer units, and retrained the entire network jointly on ImageNet and a synthetic word dataset (phase 2), which also contained 1.3 million images of 1000 words. This was considered a literate network.

We trained separate networks in 5 different languages: French, English, Chinese, Telugu, and Malayalam. To investigate the mechanisms of bilingualism [14], two additional networks were trained on bilingual stimuli: English + Chinese, and English + French. To estimate the variability across training sessions, each literate network was trained starting from the same five instances of illiterate networks. Thus, there were a total of 35 literate and 5 illiterate networks. Pytorch libraries were used to train these networks with stochastic gradient descent on a categorical cross-entropy loss. The learning rate (initial value = 0.01) was scheduled to decrease linearly with a step size of 10 and a default gamma value of 0.1. Phase 1 training lasted for 50 epochs and phase 2 training for another 30 epochs. The classification accuracy did not improve further with more epochs.

## Stimuli

To improve network performance, ImageNet images were transformed using standard operations such as "RandomResizedCrop" and "Normalize". The images were of dimension 224x224x3. To avoid cropping out some letters in the word dataset, the default scale parameter of RandomResizedCrop was changed such that 90% of the original image was retained. For fair comparisons, other operations such as flipping were not performed on the Imagenet dataset as the same operation would create mirror words in the Word dataset, which is not typical in reading.

The English and French words included frequent words of length between 3–8 letters. The Chinese words were 1–2 characters long. The Telugu and Malayalam words were 1–4 characters long, which approximates the physical length of the chosen English/French words. The synthetic dataset comprised 1300 stimuli per word for training and 50 stimuli per word for testing. These variants were created by varying position (-50 to +50 along the horizontal axis, and -30 to +30 along the vertical axis), size (30 to 70 pts), fonts, and case (for English and French). For each language, 5 different fonts were chosen: 2 for the train set, i.e. Arial and Times New Roman for English and French; FangSong and Yahei for Chinese; Nirmala and NotoSansTelugu for Telugu; and Arima and AnekMalayalam for Malayalam; and another 3 fonts for the test set, i.e. Comic Sans, Courier, and Calibri for English and French; Kaiti, simhei, and simsum for Chinese; NotoSerifTelugu, TenaliRamakrishna, and TiroTelugu for Telugu; and Nirmala, NotoSansMalayalam, and NotoSerifMalayalam for Malayalam.

The bigrams stimuli used in the dissimilarity analysis (Fig 1C–1D) were taken from earlier studies [25,26].

## Identification of word selective units

Similar to fMRI localizer analysis, a unit was identified as word selective if its responses to words were greater than the responses to nonword categories: faces, houses, bodies, and tools by 3 standard deviations. The body and house images were taken from the ImageNet dataset.

For tools, we used the "ALET" tools dataset, and Face images were taken from the "Caltech Faces 1999" dataset. We randomly chose 400-word stimuli and 100 images each from the other categories for identifying category-selective units.

### Dissimilarity measure

For each layer, we vectorized the activation values, and estimated the pair-wise dissimilarity value using correlation metric i.e., d = 1-r. where r is the correlation coefficient between any two activation vectors.

### Letter selectivity

For each word-selective unit, its tuning profile across letters was estimated by identifying the letter that evoked maximum response at either of the 8 possible positions. First, we obtained 26x8 = 208 responses, followed by the max operation across positions, and were further sorted to identify the most and least preferred letter for a given unit.

### Encoding model

To predict the variability in response profile across 1000-words for word selective units in the avgIT layer, we trained a linear encoding model. Each word was represented using a vector of length 208 (26 letters x 8 positions). This vector comprised of 1s and 0s indicating the presence of a letter at a given position. To estimate the features that activate a given unit, we solved the linear equation Y = Xb using cross-validated regularized linear regression (LassoCV). Here, Y is a 1000x1 vector corresponding to the activation of a given unit across 1000 words presented at the center, X is a 1000x208 feature matrix, and b is a 208x1 vector of unknown weights.

In this study, we compared the model fits across the following three position schemes: Left aligned, word centered, and edge aligned (Table 1). Since the number of parameters is identical across these three model types, we used correlation coefficient as a metric to identify the position scheme encoded in these networks.

### Position tuning

To estimate the overall specificity of the neural responses to a given position, we first reshaped the estimated model coefficients into a 26x8 matrix and then averaged across the first dimension. This resulted in a 1x8 vector for each word selective unit that were then grouped based on the position of the peak value within this vector. Finally, the position tuning profile was estimated by averaging across units within each group.

### Response profile of word selective units

To avoid handpicking units and to quantify the global progression from retinotopic to ordinal position coding, we used an image processing-based approach to categorize each unit's

**Table 1. Position of individual letters based upon different encoding schemes.**

| Position scheme | Letter position | | | | | | | |
|---|---|---|---|---|---|---|---|---|
| | 1 | 2 | 3 | 4 | 5 | 6 | 7 | 8 |
| Left Aligned | W | O | R | D | | | | |
| Word centered | | | W | O | R | D | | |
| Edge Aligned | W | O | | | | | R | D |

response pattern as row, column, diagonal, or mixed. Specifically, we analyzed each 5x4 response matrix (e.g., Fig 3C and 3D) as follows:

1. We transformed the matrix to binary elements. The threshold was set at 30% of the difference between the maximum and minimum values of that matrix.

2. Units with a maximum response of less than 5 were excluded from the analysis.

3. We developed custom scripts to categorize each binary matrix as word position selective (row structure), ordinal letter position (column structure), retinotopic letter position (diagonal structure), or mixed selectivity (mixed structure). Specifically, we looped through each element of the matrix, comparing it with adjacent elements in different directions. Thereby counting consecutive identical non-zero elements in rows, columns, diagonals, and reverse diagonals. A unit is considered to encode ordinal position only if it has non-zero entries for the column.

This systematic approach allowed us to quantify the transition from retinotopic to ordinal coding across layers without any subjective bias.

## Connectivity between V4 and IT units

Each layer in a CNN undergoes two stages of processing: Convolution and Max Pooling. Here, the convolution filters have a kernel size of 3x3, along with padding and a stride of 1. Consequently, the dimensions of the output matrix after the convolution operation remain unchanged. The Maxpool2d operation also employs a 3x3 kernel with a stride of 2, which reduces the dimension of the output matrix to half of the input. In our network, the dimensions of the V4 layer are 14x14x256. In this context, 14x14 represents the spatial position, and there are 256 distinct features extracted at any given spatial position (256 "filters"). Each IT unit is therefore connected to the V4 layer through 256 weight matrices, each of which has a dimension of 3x3. Since these weights are applied to outputs that have undergone two stages of processing, the effective receptive field of an IT unit relative to the V4 layer is 5x5.

To characterize the functional connectivity of a given IT unit from V4 units, we began by summing the weights of each convolution filter, resulting in a 256-length vector. We opted for this summation approach instead of using L2norm or any other absolute measure, primarily because the ReLU operation nullifies all negative responses, ensuring that negative weights only dampen activity. Next, within the 5x5 receptive field of an IT unit, we identified word-selective units and assessed their responses to the preferred stimulus of the chosen IT unit. The stimulus set consisted of 20 stimuli created by combining a single preferred letter and 3 copies of the least preferred letter at various ordinal positions (Fig 3A). For each word-selective unit in V4, the maximum response across those 20 images was recorded. To obtain a single input activation value for each of the 256 V4 filters, we performed the max operation on all word-selective units for a given filter, resulting in another set of 256-length vectors. The product of the input V4 activation with the weight vector enabled us to distinguish between features that either activate or inhibit the response of the selected IT unit. The top two V4-to-IT connectivity filters that most significantly contribute to its response are shown in Fig 5. We performed a similar analysis to estimate the connectivity between V2 and V4 units.

## Activation maximization

We used the Lucent toolbox (https://github.com/greentfrapp/lucent) to generate images that maximally activate a given unit. Given the convolutional structure of the network, all units within a given channel have the same features but at different spatial locations. Thus, we only

estimated the preferred input for the unit whose receptive field was at the center of the image. For example, the output from the IT layer is a 512x7x7 tensor, where there are 512 channels, and each channel contains a 7x7 grid of units that span the entire stimulus. First, we identify the unit with the highest activity, which can belong to any of the 512 channels and any of the 7x7 receptive field positions. For consistency, we then choose the center position within that channel for our visualization. We generated images that maximized activation in both positive (Fig 6) and negative (Fig J in S1 Text) directions with a threshold of 1000 iterations.

## Supporting information

**S1 Text.** Fig A: Discriminating languages with same script. Fig B: Representation space span for letters. Fig C: Examples of units with mixed selectivity. Section A: Connectivity between IT and V4 units. Fig D: Functional connectivity between V4 and IT units (edge position tuning). Fig E: Functional connectivity between V4 and IT units (mid position tuning). Section B: Connectivity between V4 and V2 units. Fig F: Functional connectivity between V2 and V4 units (edge position tuning). Fig G: Functional connectivity between V2 and V4 units (mid position tuning). Section C: Emergence of blank space coding units. Fig H: Visualization of V1 filters. Fig I: Functional connectivity between V1 and V2 layers. Fig J: Activation-maximization of word selective units (negative direction).
(DOCX)

## Acknowledgments

This work was granted access to the High-Performance Computing resources of the Institute for Development and Resources in Intensive Scientific Computing under Allocation 2021-AD011012288 made by the Grand Equipement National de Calcul Intensif.

## Author Contributions

**Conceptualization:** Aakash Agrawal, Stanislas Dehaene.

**Data curation:** Aakash Agrawal.

**Formal analysis:** Aakash Agrawal.

**Funding acquisition:** Aakash Agrawal, Stanislas Dehaene.

**Investigation:** Aakash Agrawal.

**Methodology:** Aakash Agrawal, Stanislas Dehaene.

**Project administration:** Stanislas Dehaene.

**Resources:** Stanislas Dehaene.

**Software:** Aakash Agrawal.

**Supervision:** Stanislas Dehaene.

**Validation:** Aakash Agrawal.

**Visualization:** Aakash Agrawal.

**Writing – original draft:** Aakash Agrawal.

**Writing – review & editing:** Aakash Agrawal, Stanislas Dehaene.

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
