## [Decision Letter · Decision Letter 0]

27 May 2024

Dear Dr. Agrawal,

Thank you very much for submitting your manuscript "Cracking the neural code for word recognition in convolutional neural networks" for consideration at PLOS Computational Biology.

As with all papers reviewed by the journal, your manuscript was reviewed by members of the editorial board and by several independent reviewers. In light of the reviews (below this email), we would like to invite the resubmission of a significantly-revised version that takes into account the reviewers' comments.

We cannot make any decision about publication until we have seen the revised manuscript and your response to the reviewers' comments. Your revised manuscript is also likely to be sent to reviewers for further evaluation.

Sincerely,

Tim Christian Kietzmann, Dr. rer. nat.

Academic Editor

PLOS Computational Biology

Andrea E. Martin

Section Editor

PLOS Computational Biology

Reviewer's Responses to Questions

**Comments to the Authors:**

Reviewer #1: Attached

Reviewer #2: The review is uploaded as attachment as well.

Reviewer’s summary

This work approaches the question how an invariant ordinal code for letter strings is achieved by neural circuits using convolutional neural networks. The authors investigate the unit’s properties of several literate networks and discuss how they align with human word recognition. Furthermore, a new coding principle is discovered, that integrates ‘spaces’ in the positional encoding of letters. The results tie together into a final understanding of how low-level information is combined by learnt features to achieve invariant word recognition. The novelty of this work is the use of CNNs to investigate the emergence of codes for written words without hand-crafting features (as was common in cognitive models).

Introduction

1 L71-74 “Here, we show that precise predictions about the neurophysiological architecture for invariant reading can be obtained by studying convolutional neural networks (CNN) models of the ventral visual cortex that, like literate humans, can be recycled for reading.”

If the term 'recycling' in this context refers to 'neural recycling, it would be good to rephrase the sentence such that the term is applied to neural representations. Now it ambiguously implies that entire networks or literate humans are recycled.

Results

Invariant word identification.

2.1 When reading, I noticed I really required more information on the datasets and training to understand what you did in this section. The methods section provides sufficient details, but I would try to be a bit more explicit here in the results section. For example, it is not clear from the description in L83 whether you keep training on the image data once the 1000 words are introduced, while this is an important detail.

In addition, it is not directly clear what kind of image categories you are training on, as technically, the images of words are also image categories. This could be easily resolved by mentioning the dataset earlier.

Lastly, it seems from the text like you train several network instances on all languages at once (inspecting figure 1 and the Methods clarify that that is not the case). Perhaps you could differentiate between mono- and bilingual networks? In any case, I would recommend clarifying which networks you train on which data more clearly in the text.

2.2 From (the caption of) figure 1A, it is not clear that the literate network is trained in two phases.

2.3 How are the lengths of the two training phases determined?

Emergence of script-specific units.

3.1 L103-104. “Although few, there were units in each network that exhibited selectivity to novel scripts but not the trained script” What does the selectivity tuning of these units look like? I see you describe how similarity between scripts influences these responses. It would be interesting to know if scripts are more likely to activate the same subnetworks and features in earlier layers of the CNN if scripts are more similar with regards to their low-level features. Perhaps you can these results together with your activation maximization analysis, in which you only consider the monolingual French network?

3.2 L113-114. Is it possible to distinguish two languages that use the same script in the network representations (e.g., French and English) or does the network treat them as one?

3.3 Figure 1B. How would you explain the difference in English-selective units between English+Chinese and English+French? In this plot it is unclear how many units are selective to multiple languages, so a unit can be counted multiple times, is that correct? i.e. when inspecting the total number of units selective to French AND English, would they count up to the same number for the overlapping units in the English + Chinese and the English + French model? Relatedly, I observe that the number of French and English units is always roughly the same in each network, do you know why?

Improvements in neural discriminability.

4.1 L134. As you mention, perceived dissimilarity between letters, and not only bigrams and words, increases with literacy. It would thus be interesting in seeing if this effect applies to your networks as well. This could give more insight into the granularity at which it is learning orthographic features.

4.2.1 Related: It is not entirely clear why you conduct the bigram comparison in the French network only (versus illiterate network), but the word comparison between two literate networks. To match your claims about similarity to human neural responses, this could benefit from better control analyses.

4.2.2 The caption of 1D says the dissimilarity is calculated between bigram pairs, while the text (L139-140) seems to indicate the analysis is between words?

4.3 L145. How many unique symbols do the other languages have, and how does it impact your analyses?

Letter tuning.

5.1 L154-158. Human readers show sensitivity to edge letters. It is interesting that you find such sharp position tuning and sensitivity to the beginning and ending of sentences in a feedforward network too. In humans, this effect is also connected to the linear and temporal aspect of reading, which the CNN does not have. Do you think it has to do with higher contrast at the edges / “crowding” in the centre?

5.2 Figure 2A. It is rarely mentioned how/in what unit you measure selectivity, that would be helpful to mention in the caption for this figure specifically.

5.3 Figure 2C is a bit confusing and crowded (although the caption and methods clarify well). I recommend giving it a more informative title than ‘mean normalized coefficient’ (and put that as axis label instead).

Emergence of letter- and position-invariant units.

6.1 Figure 3:

6.1.1 A small detail that does not require change: flipping the axes of these matrices would be more intuitive to me (absolute word position on the x-axis, letter ordinal position as y-axis.

6.1.2 Axis labels would be helpful in 3B, 3C, and 3D too.

6.1.3 For clarification on figure 3B: you show 2 types of units, but 4 units (hence 4 colours)?

6.2 It is great to see a few example units, but it would be helpful to get a global measure to inspect the progression from early to later layers, (which you mention in the text as percentages). Relatedly, is there a way to quantify the transition from retinotopic to ordinal, or do you have to handpick these units?

Emergence of ordinal position coding units

7.1 Very nice analysis. It would be helpful to mention here explicitly that you choose V4 and IT because that is where you observe the transition from retinotopic to ordinal codes.

Optimal stimuli for each layer

8.1 You wrote that all sections after Figure 1 would be performed on the French network, but you are using English word stimuli in Figure 6. Are you using the English network here instead?

8.2 L323-326 “Remarkably, from IT and output, the automated image optimization process recovered word fragments that partially matched the word identity originally used to select the units” This is interesting, and I am curious why. Is the repetition of the word in the visualisation explained by weight sharing upstream?

8.3 The methods of visualization and unit selection are not entirely clear to me, and I have several questions about them. In the caption of Figure 6 you write: “[..] the features of the channel whose units evoked the highest response within a given layer are shown. For visualization purposes, features are displayed at the central location” And in Methods: “Given the convolutional structure of the early stages of the network, all units within a given channel have the same features but at different spatial locations. Thus, we only estimated the preferred input for the unit whose receptive field was at the center of the image.”

8.3.1 First, you mention the early stages of the network, but it’s only the H layer and readout that are not convolutional, is that correct?

8.3.2 Secondly, “displaying the features in the centre” sounds like you are moving the visualization. If you are performing this analysis on a centre unit, which has a central RF, why do you need to move the visualisation?

8.3.3 The first sentence in the figure caption, which describes that you select the channel with the highest response in the layer, does not seem to match with the methods section where you write that you select the unit in the centre position for this analysis. Are you selecting the n_channel units that have the centre position (share RF) in the 3D layer, and from those, selecting the unit with the highest activation?

8.3.4 In the methods you write: “For ease of visualization, only the images generated along the positive directions are shown.” Do the images in the negative direction show anything interesting? If so, this could be added to supplement.

Methods

9.1 The first section writes: “IT, avgpool IT or H”. I did not take away easily from your paper that H is the average pooling layer before the linear readout. It could be better described or visualised in architectural figure 1, that this is the pooling of IT with a new name. Additionally, since the other layers have names corresponding to visual cortex, it would be helpful to know the rationale behind naming this final layer 'H', and why you use it in your first analyses, and not IT. It would thus be great if you can include a clearer description of the layer choices, more details about the architecture (for example by adding a table with all layers and respective parameter details), and finally a justification of the use of the H layer in the first few analyses.

9.2 With regards to the stimuli: the random flipping data augmentation was also not used training the illiterate network on Imagenet?

Minor details

10.1 L167-177: “each unit’s receptive [field]”

**Have the authors made all data and (if applicable) computational code underlying the findings in their manuscript fully available?**

Reviewer #1: Yes

Reviewer #2: **No: **I could neither find a reference to the code repository in the manuscript, nor find it after searching online.

PLOS authors have the option to publish the peer review history of their article (what does this mean?). If published, this will include your full peer review and any attached files.

Reviewer #1: **Yes: **Jeffrey Bowers

Reviewer #2: **Yes: **Victoria Bosch
---

## [Decision Letter · Decision Letter 1]

19 Aug 2024

Dear Dr. Agrawal,

We are pleased to inform you that your manuscript ' Cracking the neural code for word recognition in convolutional neural networks ' has been provisionally accepted for publication in PLOS Computational Biology.

Best regards,

Tim Christian Kietzmann, Dr. rer. nat.

Academic Editor

PLOS Computational Biology

Andrea E. Martin

Section Editor

PLOS Computational Biology

Reviewer's Responses to Questions

**Comments to the Authors:**

Reviewer #1: I am pleased with the authors responses and happy to recommend accept. Jeffrey Bowers

Reviewer #2: The authors have effectively addressed all my concerns and questions through their revisions, which have

resulted in a significantly improved manuscript. The text, methodology, and figures are improved and clearer. I believe the manuscript is now ready for publication.

**Have the authors made all data and (if applicable) computational code underlying the findings in their manuscript fully available?**

Reviewer #1: Yes

Reviewer #2: Yes

PLOS authors have the option to publish the peer review history of their article (what does this mean?). If published, this will include your full peer review and any attached files.

Reviewer #1: **Yes: **Jeffrey Bowers

Reviewer #2: **Yes: **Victoria Bosch

---

## [Editor Report · Acceptance letter]

2 Sep 2024

PCOMPBIOL-D-24-00491R1 

 Cracking the neural code for word recognition in convolutional neural networks 

Dear Dr Agrawal,

I am pleased to inform you that your manuscript has been formally accepted for publication in PLOS Computational Biology. Your manuscript is now with our production department and you will be notified of the publication date in due course.

With kind regards,

Anita Estes
